# Naturally and Anthropogenically Induced *Lingulodinium polyedra* Dinoflagellate Red Tides in the Galician Rias (NW Iberian Peninsula)

**DOI:** 10.3390/toxins16060280

**Published:** 2024-06-19

**Authors:** Ricardo Prego, Roberto Bao, Manuel Varela, Rafael Carballeira

**Affiliations:** 1Instituto de Investigaciones Marinas (IIM-CSIC), Eduardo Cabello, 6, 36208 Vigo, Spain; prego@iim.csic.es; 2GRICA Group, Centro Interdisciplinar de Química e Bioloxía (CICA), Universidade da Coruña, Rúa As Carballeiras, 15071 A Coruña, Spain; roberto.bao@udc.es; 3Centro Oceanográfico de A Coruña, Instituto Español de Oceanografía, Apdo. 130, E15080 A Coruña, Spain; manuel.varela@co.ieo.es; 4Institut Cavanilles de Biodiversitat i Biologia Evolutiva (ICBiBE), Universitat de València, C/Catedràtic José Beltrán 2, 46980 Paterna, Spain

**Keywords:** harmful algal bloom, oceanography, bottom-up regulation, cysts, dinoflagellates, human impact, dredging, Galicia

## Abstract

Despite the fact that the first red tide reported on the coasts of the Iberian Peninsula was due to *Lingulodinium polyedra*, knowledge about their frequency and, particularly, about the environmental conditions contributing to bloom initiation is still scarce. For this reason, *L. polyedra* bloom episodes were observed and studied in three Galician rias during the summer season based on the 1993–2008 record database period; additionally, samples were collected in summer 2008. Proliferations of *L. polyedra* occurred in the rias of Ares and Barqueiro in June and August, respectively, while in the Ria of Coruña, they persisted from the end of June to early September. Red tides developed when the surface temperature reached 17 °C, with “seasonal thermal window” conditions, and when salinities were ≥30, i.e., an “optimal salinity window”; when these parameters were lower than these thresholds, cyst germination decreased. A cyst transport mechanism from sediments to the surface must also exist; this mechanism was found to be natural (tidal currents) in the ria of Barqueiro or anthropogenic (dredging) in the rias of Ares and Coruña. Surface temperatures during summer were usually favorable for cyst germination (85 to 100%) during the 1993–2008 period; however, water temperatures below 10 m depth only rarely reached the 17 °C threshold (2 to 18%). During this 16-year period, dredging activities could explain 71% (Coruña) and 44% (Ares) of the recorded bloom events. When a bloom episode developed in early summer, favorable conditions did not lead to a new red tide, probably due to the lag period required by cysts for germination. Moreover, blooms did not develop when high densities of diatoms (>1,000,000 cells·L^−1^) remained in the water column as a result of summer upwelling pulses occurring in specific years. The temperature–sediment disturbance pattern found in this study provides a useful tool for the prevention of eventual risks resulting from red tides of this dinoflagellate.

## 1. Introduction

High biomass microalgal blooms in coastal areas is a phenomenon by which phytoplankton concentration rapidly increases. Among these blooms, those composed of dinoflagellates are among the most impressive due to the reddish color they can give to the water. These blooms can spread over large areas and are commonly known as “red tides”. The fascination that this singular group has always caused [1,2,3] goes beyond the purely academic interest, due to the toxic nature of many red tides [4]. Harmful algal blooms (HABs) caused by dinoflagellates, although their causes are still poorly understood, have a great ecological impact on marine food webs due to the ability of dinoflagellates to produce different types of toxins, which can affect fish populations. For this reason, these events can have a considerable impact on local economies and health [5,6,7,8]. A distinctive bloom-forming species is the ecologically important *Lingulodinium polyedra* (Stein) Dodge (=*Lingulodinium polyedrum* (Stein) Dodge, *Gonyalulax polyedra* Stein). *L. polyedra* is a single-cell marine photoautotrophic planktonic protist, a typical species of the dinoflagellate group [9], whose motile stage was first described by Stein in 1883 [10,11]. This dinoflagellate is considered a cosmopolitan warm-water species and has been recorded in many parts of the world [12]. However, it should be understood as a species restricted to warm temperate to tropical coasts only. Therefore, at the worldwide coastal scale, cyst germination and eventual blooms would depend on water temperature [13,14,15,16]. Interestingly, *L. polyedra* has become a highly frequent bloom-forming species in the temperate European Atlantic coasts, such as those of the Iberian Peninsula, in recent years [16,17,18,19].

Reports concerning the distribution of *L. polyedra* indicate that both the motile and the cyst stage mainly occur in estuaries and other coastal systems [9], such as the Galician rias, located in the northwestern corner of the Iberian Peninsula [20]. A few years after the first description of a *L. polyedra* red tide, which occurred in the Californian coasts in July 1902 [21], Sobrino [11] identified and carried out a detailed morphological description, including microscopic photographs, of this dinoflagellate collected from a red tide in the ria of Pontevedra in summer 1917 for the first time in the Iberian Peninsula coasts [22]. The ecology of *L. polyedra* in the rias was not addressed until the 1950s, when Margalef [23] mentioned its presence in the ria of Vigo. Subsequently, Margalef [24], based on samples collected in August and September 1955, described the oceanography and organisms associated with red tides. He found concentrations of up to 2,180,000 cells·L^−1^ of *L. polyedra*. Surprisingly, until the 1950s, *L. polyedra* was the species recorded as causing most blooms and seawater discoloration episodes in the Galician rias located to the south of Cape Finisterre. Since then, no red tides of *L. polyedra* were reported in the Galician coasts, despite the large number of cysts present in sediments from some of those rias [25,26], until two episodes occurred north of Cape Finisterre: a red tide in the ria of Ares in 2008 [27] and a bloom event in the ria of Barqueiro in 2011 [28].

Often, the final phase or step of a red tide is the formation of cysts, which act as a resistant resting stage that remains in sediments until conditions are again favorable for growth [1]. Their permanence in anoxic conditions in sediments seems to be important for the viability of dormant cysts and the development of red tides, particularly of *L. polyedra* [12]. Many studies have tried to establish the role of cysts in triggering algal blooms, and the theory of seed populations as the main cause of red tide development has been postulated [29,30]. Thus, sediment disturbance could play a role in promoting the onset or increase of algal bloom events of *L. polyedra*.

In Galicia, red tides have become an increasing matter of concern, as mussel farming using rafts inside the rias, introduced in the 1950s, has become a major industry, yielding important economic benefits. Since 1976, these events have been recognized as potentially toxic. In several parts of the world, *L. polyedra* has been associated with the production of paralytic toxins, such as yessotoxins [31,32], and ichthyotoxins [33]. However, no associated toxicity has been reported for most red tide episodes of this species. In particular, this species has not been related to toxic episodes in the Galician rias, even though its toxins have been detected during a bloom of this species in the ria of Ares [34]. In this ria, yessotoxin concentrations in a mussel raft area were found to be below the regulatory threshold required by the Galician monitoring system [35].

Although the first red tide reported in the coasts of the Iberian Peninsula was due to *L. polyedra*, and despite the potential toxicity of this species, the available knowledge about its frequency and, particularly, about the environmental conditions contributing to triggering blooms, is still scarce. Our hypothesis is that anthropogenic disturbance of sediments, together with suitable temperatures, plays an important role in triggering red tides of *L. polyedra*. To test this, the presence of bloom events in the Galician rias was studied with the following objectives: (i) to describe the temporal variability in the abundances of *L. polyedra* in three rias located to the north of Cape Finisterre during summer 2008; (ii) to determine the environmental variables triggering the development of blooms, paying special attention to water temperature and sediment disturbance, and (iii) to determine the importance of dredging activities, as a vector for cyst transport and seeding from the sea bottom to the upper layers of the water column, in inducing red tides, based on historical data from the rias of Coruña and Ares.

## 2. Study Area

The shoreline of Galicia (a region with 1720 km of coastline in the northwest of Iberian Peninsula) is a ria-coast system, where the larger rias are located to the south of Cape Finisterre, while the smaller ones, such as the rias of Coruña, Ares, and Barqueiro (Figure 1), are located to the north. These three rias can be considered funnel-like incised valleys characteristic of a relatively submerged coastline with around 30 m depth at their mouths, and open to swell from the north or northwest. Between 10% (in the case of the ria of Coruña) and 25% (in the remaining rias) of their area consists of intertidal flats with a mesotidal regime. This geographical region has a wet temperate oceanic climate (Cfb Köppen type) [36].

The ria of Coruña (16 km^2^) has a large oceanic influence, with the exception of its innermost zone, into which the Mero River flows (8.2 m^3^·s^−1^ of average annual flow), and its salinity shows fluctuations similar to those of an estuary [37]. Anthropogenic alterations in morphology (harbor jetty) and freshwater discharge (subjected to the administration of a river dam) resulted in significant changes in the circulation pattern of the ria [38]. In the enclosed zone of the inner harbor, turnover is very low, resulting in the retention of phytoplankton, which shows a high biomass (with mean chlorophyll ranging from 5 to 17 mg·m^−3^) dominated by large diatoms, with a negligible contribution of flagellates, except during downwelling and winter mixing periods [39].

The ria of Ares (72 km^2^) has an innermost zone divided into two estuarine branches, where the River’s Eume (16.5 m^3^·s^−1^) and Mandeo (14.1 m^3^·s^−1^) (Figure 1) flow into. The middle-outermost ria, as in Coruña, behaves like an extension of the adjacent shelf, and a hundred mussel rafts are anchored in the Lorbé cove. Bode and Varela [40] showed a differential upwelling effect in the Ria of Ares compared to the other rias located to the south of Finisterre due to the dissimilar distribution of upwelling in the region [41].

Several dredging activities have been conducted in the last 20 years in the rias of Coruña and Ares (near stations C3, A3, and A4; Figure 1). Information on dredging activities in these rias, collected from both the harbor authorities of Coruña and Sada and from the companies that carried out works causing sediment perturbation, is summarized in Table 1.

Specifically, from July to September 2008, port dredging activities took place on a daily basis in the Coruña harbor (post anchoring, preceded by drilling of the seabed to about 5 m, to support wharfs) to build new nautical facilities. Moreover, during this period, dredging activities in the Sada port, located in the Ria of Ares, were visually confirmed by the crew of RV Lura during weekly routine sampling in the area. For the Ria of Ares, other activities that can potentially disturb sediments must also be taken into account, such as the installation of salmon farming cages or mussel raft maintenance tasks (replacing anchoring devices or mooring new ones, etc.).

The Ria of Barqueiro (13 km^2^) is located near Cape Estaca de Bares, the northernmost point in the Iberian Peninsula. This ria is partially enclosed by well-developed beach barriers (Figure 1). Hydrodynamics is mainly defined by marine processes, except in the innermost estuarine zone, under the influence of the Sor River (15.2 m^3^·s^−1^), where the maximum chlorophyll values were observed. A mesotrophic pattern characterizes the inner ria, as revealed by nutrient and chlorophyll concentrations and plankton abundances [28]. No dredging or any other sediment-disturbing human activities were observed throughout the year 2008 in the Ria of Barqueiro.

## 3. Results

Counts of *L. polyedra* in surface water samples from the three rias showed that blooms of this species occur essentially during summer. During the remaining seasons, *L. polyedra* was either completely absent or present in negligible concentrations. For this reason, we focused on the period extending from July to September.

### 3.1. Red tides during Summer 2008

#### 3.1.1. Thermohaline Variables

During summer 2008 in the Ria of Coruña (Sts. C1–C3) (Figure 1), a continuous temperature and salinity gradient from the harbor towards the adjacent shelf area was observed. These hydrographical parameters ranged from 15 to 20 °C and salinity from 34.2 to 35.7, respectively. The lowest surface temperatures, in the range of 15–19 °C, were recorded in shelf waters (St. C1), whereas the Coruña harbor exhibited temperatures between 16.5 to 20.1 °C during the same period (St. C3). In contrast, salinity exhibited an opposite pattern with the aforementioned lower range. Inside the Ria of Coruña, a vertical thermal gradient was observed. Up to a depth of 5 m, temperature values were similar to those on the surface, but temperatures at depths greater than 5 m remained below 17 °C.

Surface data (0–5 m depth) for thermohaline variables during summer 2008 in the stations of the Ria of Ares (St. A1–A4 in Figure 1) showed salinities typical of the seawater, ranging from 34.6 to 35.5, except for St. L4 (34.0–35.3). Regarding temperature, they were always higher than 17 °C in the four stations (17.1–20.1 °C) [42,43].

In the Ria of Barqueiro (Sts. B1 and B2) (Figure 1), surface temperatures recorded during samplings on July, August, and September 2008 ranged from 19 to 21 °C in both stations (18.8–19.9 °C in B1 and 17.8–19.0 °C in B2 during one tidal cycle), whereas bottom water temperatures ranged from 14.5 to 18.9 °C (St. B1). Large differences were found for other variables. In July, salinity ranged between 20 (low tide) and 33 (high tide) in St. B2, increasing to a range of 29–35 in August and 27–35 in September. A similar trend, although with a narrower range of variation, was recorded in St. B1, where salinity increased from 34.7 (July) to 35.5 (August and September). In St. B2, tidal current velocities also increased from July (12 and 0.5 cm·s^−1^ at low and high tide, respectively) to August (68 and 18 cm·s^−1^) and September (29 and 26 cm·s^−1^). Average tidal current velocity varied from 35 (July) to 67 (August) and 74 (September) as a result of tidal amplitudes of 2.3, 3.6, and 3.8 m, respectively.

#### 3.1.2. Abundances of *L. polyedra*

Temporal variations over 2–3-day intervals at St. C3 in the Coruña harbor showed cell density values up to 300,000 cells·L^−1^ (Figure 2A). Algal blooms persisted for two months, coinciding with the construction of harbor facilities. Visual observations showed a lack of uniformity in the distribution of *L. polyedra* spots when the water color turned reddish. The sampling station usually appeared with a less intense color than other harbor areas.

In the Ria of Ares, red tides also showed a shorter persistence than in the ria of Coruña, although cell concentrations were higher (up to 700,000 cells·L^−1^). Figure 2B shows the mean values corresponding to the four stations, A1 to A4, distributed throughout the ria. The dispersion of red tides was not uniform, with variable values among stations. In this area, algal blooms were also observed in June.

The estuary in the innermost zone of the Ria of Barqueiro showed maximum values in August, with abundances around 50,000 cells·L^−1^ (Figure 2C). The species was also observed in the middle section of the ria (St. B1), but abundances were lower than 10,000 cells·L^−1^. This density could be insufficient for this event to be considered as a red tide, as also suggested by the lack of reddish coloration in the water. However, considering the average abundances of phytoplankton over an annual cycle in this ria, the above-mentioned densities are relevant [28]. As samples in the ria of Barqueiro were obtained only during full moon periods, it was not possible to follow the daily or weekly evolution of algal blooms, as in the rias of Ares and Coruña. Nevertheless, these summer results are useful for comparison purposes with variations in the occurrence of the target species in other areas.

### 3.2. Red Tides during Summer from the Period 1993 to 2008

Figure 3A summarizes all the available information on dredging activities, red tides (cell abundances of *L. polyedra*) in the harbor area, and bottom water temperatures (at a depth of 20 m) in the Ria of Coruña during the 1997–2008 period. No data on *L. polyedra* were available for the year 2000, and other information was not found on the occurrence of red tides in the ria.

During the years 2001 and 2008, dredging activities took place simultaneously to red tide events reaching concentrations >100,000 cells·L^−1^ (Figure 3A). Bottom water temperatures during these events were lower than 17 °C, but surface temperatures always exceeded 17 °C [42,43]. In 1999, high temperatures of nearly 17 °C were recorded at the bottom; they were simultaneous with the occurrence of moderate *L. polyedra* abundances around 50,000 cells·L^−1^ (Figure 3A). In summer 2003, a red tide event with a density of 400,000 cells·L^−1^ (Figure 3A) developed simultaneously with bottom water temperatures exceeding 17 °C.

In the Ria of Ares (Figure 3B), red tides were a common occurrence (with concentrations up to 2,500,000 cells·L^−1^; see Figure 3B). Several blooms coincided with dredging activities, e.g., in 1993–1994, 2000–2002, and 2008. No data on *L. polyedra* abundances were available for the 2005–2007 period. Assuming similar water temperature values during the bloom events in this ria to those occurring in the ria of Coruña, they would always have exceeded 17 °C. Also, red tides in 1996 and 1999 up to 100,000 cells·L^−1^ (Figure 3B) and 2003 and 2004 up to 2,500,000 and 600,000 cells·L^−1^, respectively (Figure 3A) were coincident in the ria of Coruña with bottom water temperatures higher than 17 °C. In addition, there is a significant correlation between *L. polyedra* cell densities with dredging activities (*p* = 0.670; *p*-value < 0.05) and temperature of water in bottom > 17 °C (*p* = 0.580; *p*-value < 0.05) in both rias.

## 4. Discussion

### 4.1. Environmental Conditions Related to Bloom Events of Lingulodinium polyedra

The coastal areas of the Artabro Gulf have been routinely monitored since the beginning of the 1990s. No red tides had been observed in the Ria of Coruña before 2001 (Figure 3). Some papers [44,45] even considered the absence of red tides as a peculiarity of this area. However, in September 2001, in the Coruña harbor, a bloom of *L. polyedra* around ≈ 400,000 cells·L^−1^ (Figure 3) occurred coincidentally with dredging operations [46,47], suggesting that sediment disturbance could very likely be the mechanism triggering the initiation of red tides. In this context, sediment resuspension brings cysts to the upper layers of the water column, where environmental conditions have to be adequate for excystment. Cysts have been related to the development of red tides [9,48]. Thus, these events are more likely to occur in areas with higher abundances of cysts in sediments. So far, no data are available on cysts in sediments of the Ria of Coruña; however, Blanco [25,26] reported the existence of high abundances of cysts in sediments of the ria of Ares and, which could be extended to the Ria of Coruña due to their proximity.

Thermohaline data from the Ria of Coruña have been used to define an environmental temperature and salinity window for the presence and blooms of *L. polyedra* (Figure 4). Red tides developed when temperature exceeded 17 °C, reaching maximum cell abundances at temperatures between 18 and 19 °C, and salinity varied between 34 and 35. Salinity showed a narrower variability during the summer due to the very low river flow to this ria [49]. Temperature and salinity data available for the Sts. 1–4 (Figure 1) in the Ria of Ares in summer 2008 [42,43] were similar to the Ria of Coruña, presenting salinities and temperatures higher than 34.0 and 17 °C, respectively. The same occurs for the Ria of Barqueiro [28]. Blanco [13] suggested that temperature controls the rate of cyst germination, while Peña-Manjarrez [50] pointed out that when water temperature increases to approximately 17 °C, a “thermal window” is reached for this species, promoting cyst germination, together with a decrease of their relative abundance in surface sediments. This role of temperature was also proposed by other authors within a temperature range from 17 to 23 °C [15,50,51,52]. Maximum threshold temperature may change according to geographical area, but the minimum temperature reported for cyst germination is always 17 °C. This is the case observed in the Ria of Coruña, when below 17 °C the seawater column remains stable and stratified during summer [39] and bottom waters do not meet the temperature requirements for cyst development (Figure 3). This condition has been observed during the summer of 2008 in the rias of Coruña and Ares. Red tides of *L. polyedra* developed in both rias with a density > 100,000 cells·L^−1^ (Figure 2) associated with sediment reworking and transport to the upper layers. According to the temperature threshold previously defined based on the available data, surface temperature allowed for cyst germination, while bottom temperature did not reach the 17 °C threshold. The “thermal window” required to trigger excystment in this species [50,51] only applies to surface water. Therefore, in the stated conditions, the only way to initiate a red tide should be the transport of cysts to upper layers, with dredging acting as the transport vector (Figure 3).

During August 2008, a *L. polyedra* red tide was also observed in the innermost part of the Ria of Barqueiro, to the North of the Artabro Gulf (St. B2) (Figure 1). A clear gradient was observed, with densities decreasing seaward, confirming that the bloom was originated in the innermost area. In this ria, no dredging or other anthropogenic activities that could cause sediment disturbance took place. In summer 2008, water temperatures exceeded the 17 °C threshold (Figure 2C) needed for cyst germination and eventual red tide development. In the Ria of Barqueiro, significant differences between data collected in July and August were observed regarding three parameters. The first one was *L. polyedra* abundance: no cells of *L. polyedra* were observed in July, while high abundances at low tide were found in August, with a density of 55,000 cells·L^−1^ (Figure 2C). The second divergence concerned salinity (measured at low tide), which was 20 In July and 30 in August (Figure 2C). Culture experiments with a wide salinity range of 10–40 found that low-salinity cultures displayed poor or no growth (S < 10) [53] and significant yessotoxin cell quotas [54]. In the Galician coast, salinity values lower than 30 led to a noticeable reduction in the germination rate of *L. polyedra* cysts [12]. The third difference was observed in tidal current velocity. In station B2, in the innermost section of the Ria of Barqueiro (Figure 1), current velocity at low tide was highest in July (Figure 2), with maximum current velocities of 77 cm·s^−1^ in July and 126 cm·s^−1^ in August as a result of tidal ranges of 2.3 and 3.6 m, respectively. The influence of high current velocity due to August spring tides can lead to sediment disturbance prompting cyst resuspension. Together with temperature values > 17 °C, both factors, salinity and current velocity, were probably relevant and complementary for the initiation of the red tide event that took place in August. In any case, blooms of *L. polyedra* in the Ria of Barqueiro have an exclusively natural origin. Surprisingly, during September spring tides, no *L. polyedra* proliferation events were observed (Figure 2C), even though temperature, salinity and current speed values were similar to those of July. This could be due to the fact that cysts need a dormancy period before they are able to germinate [29]. In the case of *L. polyedra*, this period can range between 2 and 5 months [55], depending on the parental origin of the cells. Because of this dormancy period, the algal bloom of *L. polyedra* in the Ria of Barqueiro developed as a single summer pulse when temperature and salinity matched the requirements for cyst germination during the short spring tide period, when cyst resuspension is enhanced by higher current speeds. Conversely, in the rias of Coruña and Ares, the algal bloom detected during summer 2008 was a continuous process (Figure 2). Contrary to bioturbation, which is usually restricted to the uppermost sediments, where most of the cysts are still in their obligate dormancy condition, dredging activities affect the deeper layers rich in cysts with a potential to germinate [29,55]. This mechanism would explain the shorter (bioturbation in the case of Ría of Barqueiro) versus extended (dredging in the case of the rias of Ares and Coruña) blooms in the area.

The Galician estuaries also have partial thermohaline stratifications that cause temperatures to decrease at depth and keep the sediments anoxic during the late summer season [12]. However, stratification phenomena have also been shown to be one of the triggers of red tides off the coast of Galicia, local oceanographic conditions in which multiple environmental factors interact in upwelling-downwelling cycles (i.e., winds, tides, river discharges), which can create deep currents that resuspend the sediment and release the cysts into the water column [12]. On the other hand, together with sediment dredging, the temperature in the latter rias is the most relevant factor affecting cysts germination, because salinity values found in these rias during the summer are characteristic of oceanic waters [56].

Margalef considered red tides to be anomalies developing from stage 3 of the typical successional cycle in temperate seas, which should not occur under upwelling conditions, as is the case of the Ria of Vigo [24,57], where the main nutrient source is remineralization [58]. Regarding *L. polyedra* blooms, most studies have focused on nitrogen [9]. Downward migration at night searching for nutrients has been considered too simplistic an explanation, as vertical migration to depths with sufficient light can also occur during daylight [50]. This could occur in the rias of Coruña and Ares during *L. polyedra* red tides. In both rias, in the absence of upwelling, dredging activities resuspend cysts and can furthermore move remineralized nutrients from sediment up to the seawater column. Nutrient salts data in the Ria of Ares during summer 2008 [42,43] show nitrogen as the limiting nutrient, with nitrate depleted in the water column and nitrite concentrations lower than 0.2 µM, while ammonium was in the range of 0.2–1.0 µM. On the other hand, sediments could be the main nitrogen source in the inner zone of the Ria of Barqueiro during summer 2008. During spring tides, water that floods the intertidal bottom (flooding water) can remove the pore water of sediments [28]. Moreover, there is a nutrient input from the Sor River ≈35 µM of nitrate [28] that flows into the Ria of Barqueiro.

### 4.2. Lingulodinium polyedra Red Tides in the Rias of the Artabro Gulf

Recently, Rodríguez et al. [20] have established that *L. polyedra* blooms were restricted to the Rías Baixas, the Rías de Vigo and Pontevedra, until the middle of the 20th century from the historical record of red tides on the coasts of Galicia (1916–2011). However, the data provided in this study show a higher recurrence of *L. polyedra* blooms in the Rias de Ares (1993–2008), Coruña (1999–2008) and Barqueiro (2012) (Figure 2 and Figure 3), as shown by the most recent historical record (2003–2011) [20]. *L. polyedra* blooms in the historical record [20], as well as the data provided in this study (Figure 2 and Figure 3), show a strong seasonality with a higher recurrence in spring-summer, with the exception of autumn of 2003 in the Ria de Ares. After the period of this study (1993–2008) only one *L. polyedra* bloom has been detected in the historical record in the Ares estuary in spring-summer 2011 [20].

Once temperature (as a key variable) and sediment disturbance (as a transport vector) have been suggested as the triggering mechanisms to explain the recent appearance of red tides in the three studied rias in 2008, historical records of *L. polyedra* blooms observed in the rias of the Artabro Gulf (Coruña and Ares) have been reviewed to unveil the causes of the occurrence of red tides in the area (Figure 3). The rias of Coruña and Ares are similar in terms of the influence of dredging on red tides. In both rias, the percentage of favorable surface temperatures (>17 °C) from July to September for the 1989–2008 period was more than 70% and 80%, respectively. This percentage increased up to 100% when only July was considered. Favorable conditions for cysts germination therefore extend for most of the summer, i.e., there is a “seasonal window” as a result from the “thermal window”. Thus, *L. polyedra* red tides are usually observed during this season [20,50,51,59,60,61,62]. Consequently, when dredging is carried out during summer, red tides are highly likely to develop. Nevertheless, some differences between both rias are evident. In the Ria of Ares, the number of bloom events resulting from high bottom temperatures is six times higher than in the Ria of Coruña (Table 2). As mentioned above, red tide events in the Ria of Coruña had not been observed before 2001. Conversely, up to five events were observed in the Ria of Ares in the period from 1993 to 2001 [27].

The Ria of Ares has larger shallow areas with a depth lower than 5 m (18 km^2^), which include the innermost portions of the Ares and Sada inlets. *L. polyedra* cysts accumulate in silt and clay sediments in these areas [26]. These shallow areas in coastal environments of the bays associated with the fine sediment fraction have been defined as “discrete seedbeds” [50]. In the Ares estuary, the spatial distribution of *L. polyedra* cysts has been studied, concentrating in the two shallowest internal areas of the estuaries, below 10 m depth, areas of low hydrodynamism or hydrodynamic shade dominated by fine muds [62]. Cysts stored in the mud [63] remain viable for very long periods, up to nine years in the case of *L. polyedra* cysts [64], and local blooms commonly start in these areas [50]. In fact, the presence of red tide events in the Ria of Ares can be explained solely by the processes occurring in the Ares and Sada inlet areas, where dredging or maintenance activities at both harbors are more likely to take place (Figure 3). *L. polyedra* cysts that were stored in anoxic conditions had much higher germination percentages than those stored in oxic environment, which practically does not germinate [13]. In this way, the cysts buried deeper in the sediment make up important and effective reserve populations for motile phases [13]. In the Sada coast (St. 3 in Figure 1) of the Ria of Ares, Blanco [65] observed that the vertical distribution on *L. polyedra* in the sediment shows one or more subsurface maxima. In this way, the dredging activity can remove mud sediments at a greater depth, i.e., usually anoxic. There, a species such as *L. polyedra*, whose cysts are very resistant, are accumulated as permanent inoculum [65]. The highest cysts presence in sediment were in ria mud sediments, i.e., the Lorbé-Sada coast (St. 1–3 in Figure 1) with contents of 4–20 cysts·L^−1^ of sediment [62]. In the Ria of Coruña, 20% of red tide events were related only to suitable temperatures at the bottom; conversely, around 70% could be associated with dredging (Table 2) during periods when bottom water temperatures were lower than 17 °C but surface temperatures were higher than 17 °C, allowing for the germination of cysts transported by dredging from the sediment to the upper layers of the water column layers (Figure 3). Dredging and temperature explain 45% and 40%, respectively, of red tide events in the Ria of Ares (Table 2), while the remaining 15% could be linked to lack of data for temperature and/or other environmental variables, as was the case in around 10% of the bloom episodes in the Ria of Coruña.

In a few cases (excluding those for which no data on cell abundance or temperature were available), red tides were not observed, despite environmental conditions being favorable (Figure 3). In the Ria of Coruña, this situation occurred in 1999 (suitable bottom temperatures) and 2000 (dredging), while in the Ria of Ares, it was observed from 1997 to 1999 (suitable temperatures) and in 2002 (dredging). In all these cases, a well-developed diatom population (>1,000,000 cells·L^−1^ in the photic zone) was present in the rias as a consequence of persistent upwelling events throughout the summer [42,43]. Margalef [57] considered red tides to be part of the typical successional cycle in temperate seas, which should not occur under upwelling conditions [56,66]. In this scenario, despite favorable conditions for cyst germination to occurr, diatoms outcompete dinoflagellates such as *L. polyedra* [67], with higher growth rates and faster nutrient assimilation [68]. However, it is worth noting a peculiarity: on 15 July 2002, a red tide event was recorded in the Ria of Ares, even though the involved species was not *L. polyedra* but *Gonyaulax* sp., accompanied by *Prorocentrum* sp. with a density of 377,000–809,000 and 42,000–1,414,000 cells·L^−1^ in St. A1 and A3, respectively [42,43]. Therefore, a red tide did finally occur, but with different species involved.

## 5. Conclusions

Sediment disturbance, either natural or anthropogenic, is a key mechanism in the initiation of *Lingulodinium polyedra* red tides. Blooms develop when surface temperature reaches 17 °C and salinities are higher than 30. Also, a red tide would only be expected to start when cysts are transported to the upper layers; the “transport vector” can be dredging or any other type of sediment disturbance (spring tidal currents). Otherwise, bottom water temperatures of 17 °C are required to trigger a red tide. Our study in the Galician rias confirms the existence of a “thermal window” for this species, as previously proposed by other authors, which occurs during summer and can therefore be considered a “seasonal window” in these rias. Sediment disturbance explains around 70% of bloom events in the Ria of Coruña and about 45% in the Ria of Ares. This finding has important implications, because red tide initiation does not exclusively depend on favorable bottom temperature conditions. Favorable surface temperatures (which are more easily reached than favorable bottom temperatures) and the presence of a vector that transports cysts to the upper layers would trigger the process. Although the seeding nature of cysts has been pointed out by many authors, dredging as the mechanism triggering red tide events had not previously been suggested. *L. polyedra* is not considered a toxic species; however, some authors have reported mortalities of marine animals and the presence of yessotoxins in cultures and in mussels during a red tide of this species in Galician coastal waters, which questions the safety of this species respect to toxicity. The obtained results are relevant from a practical point of view to prevent the consequences of red tides, as anthropogenic activities in the coastal system are significantly increasing, therefore increasing the probability of red tide events. The historical data on *L. polyedra* red tides in the Galician rias provide a critical foundation for future research and management efforts. By understanding the key factors of bottom-up regulation that triggered harmful algal blooms, such as sediment disturbances, temperature conditions, and cyst transport vectors, we can develop predictive models and early warning systems to anticipate and mitigate future events. This knowledge informs better environmental management practices, such as timing dredging activities to avoid critical periods and monitoring ecosystem health indicators. It also supports the creation of regulatory frameworks to reduce human impacts and guides targeted research to uncover new mitigation strategies. Ultimately, these data are essential for protecting fisheries, aquaculture, and the local economy, ensuring the sustainability of these vital resources.

## 6. Material and Methods

Samples for identification and counting of specimens of *L. polyedra* were collected along the water column in nine stations in three Galician rias located north of Cape Finisterre (Figure 1).

In the ria of Coruña, water samples were obtained within the framework of the Time Series project carried out by the Spanish Institute of Oceanography (IEO) [69] in three areas: (a) the Coruña harbor, where station C3 was sampled daily since 1997; temperature, salinity and chlorophyll data were measured in situ using a Beckman 280 probe, and *L. polyedra* individuals were counted at intervals of one to three days, depending on changes observed in chlorophyll concentrations; (b) the middle section of the Ria of Coruña, where station C2, with a depth of 20 m, was monitored on a monthly basis since 1989 on board RV Lura at depths 0, 5, 10, 15, and 20 m; (c) the Artabro Gulf, offshore the Ria of Coruña, where station C1, with a depth of 80 m, was monitored on a monthly basis since 1989 on board RV Lura at depths 0, 5, 10, 20, 30, 40, and 70 m. In the aforementioned stations C2 and C3, vertical temperature and salinity profiles were recorded using a SeaBird 25 CTD. Light intensities were measured using a LI-COR sensor PAR CTD-lived. Chlorophyll and samples for phytoplankton identification were obtained using 5 L General Oceanic Niskin bottles mounted on a General Oceanics rosette. Phytoplankton samples from all stations, C1 to C3, were preserved with Lugol’s solution and kept in darkness until examination under a Nikon Eclipse inverted microscope following Uthermöhl’s technique [70].

In the Ria of Ares, stations A1 to A4 were routinely monitored from year 1993 by the Technological Institute for the Control of the Marine Environment of Galicia (INTECMAR). Water samples were collected at weekly intervals to support mussel aquaculture in the area. The above-mentioned four stations were sampled on board RV Lura within the framework of a cooperation project between the IEO and INTECMAR. One station was located close to the Sada harbor (st. A3), another one in the mouth of the ria (st. A4), and the remaining ones in the vicinity of Lorbé cove (st. A1 and A2, st. A1 being located in a mussel raft zone). Temperature and salinity data were measured using a SeaBird 25 CTD. Surface water was collected to identify and count *L. polyedra* abundances. Samples were processed following the same procedure as those from the Ria of Coruña.

In the Ria of Barqueiro, water samples were collected during spring tides in two stations in 2008 (within the Spanish CICYT research project INTERESANTE). One station was located on the estuary-ria boundary (st. B2), with a depth of 3 m, while the other one was located in the middle section of the ria (st. B1), with a depth of 21 m. Sample collection in st. B1 (at depths 0, 5, 10, and 20 m) was carried out on board RV Lura. Temperature and salinity data were measured using a SeaBird 25 CTD, and water samples to study phytoplankton species composition were obtained using 5 L General Oceanic Niskin bottles attached to a General Oceanics rosette. A small boat was used to sample station B2 at surface; temperature and salinity was measured in situ using a Beckman 280 probe at high and low tide. Additionally, current velocities were also measured with a Valeport 34500 current meter. Samples for *L. polyedra* counting were processed following the same methods described above for other study sites.

The relationship between the dredging activities, temperature, and cell density of *L. polyedra* in the rias was explored using Pearson correlation analysis (P), and by applying Student’s *t*-test to determine the level of statistical significance of correlations between parameters. The statistical analyses were performed using XLSTAT [71].

## Figures and Tables

**Figure 1 toxins-16-00280-f001:**
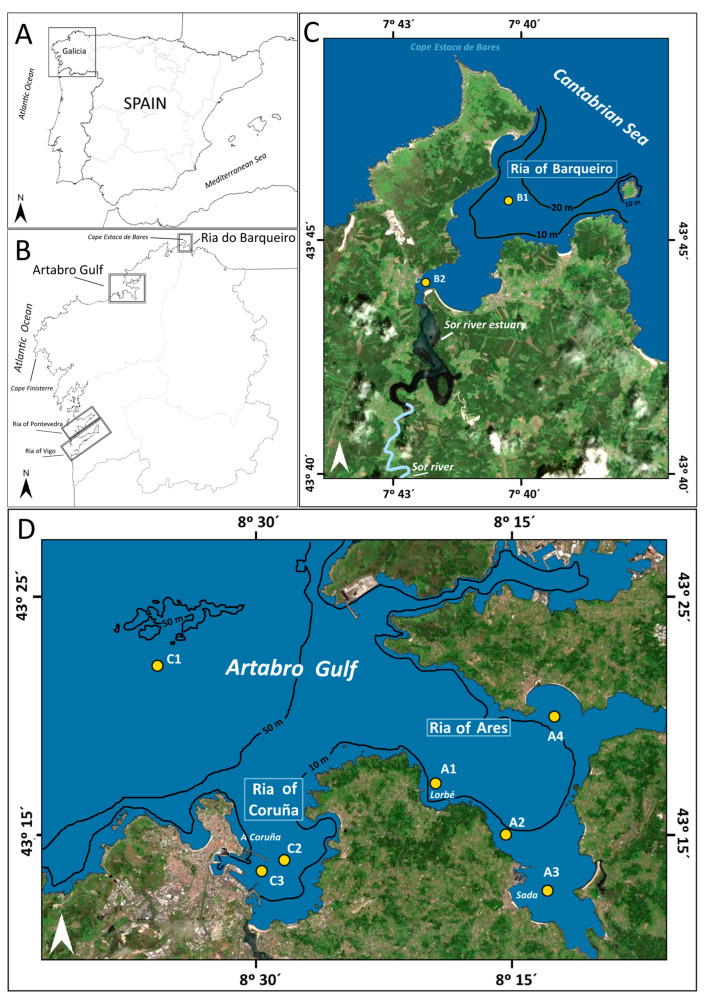
Situation map of the (**A**,**B**) study area in northern Galician rias (NW Iberia) and station locations where the *L. polyedra* abundances were quantified in the (**C**) Ria of Barqueiro (B1–B2), and (**D**) Rias of Ares (A1–A4) and Coruña (C1–C3).

**Figure 2 toxins-16-00280-f002:**
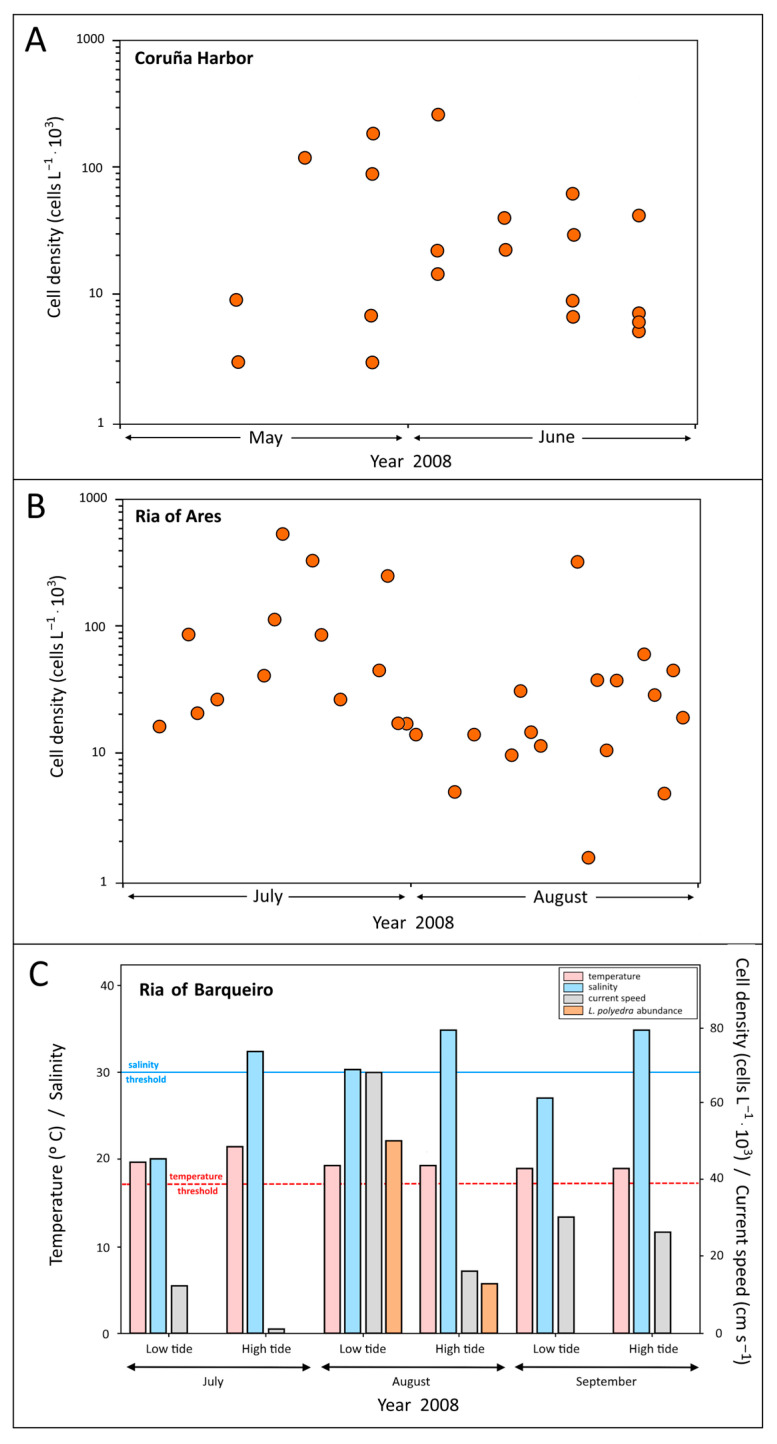
Temporal variation of *L. polyedra* abundance in (**A**) Coruña harbor, (**B**) Ria of Ares, where dots represent the individual values counted for each day and station during the period of study, and (**C**) Ria of Barqueiro showing temperature (t), salinity (s), current speed (cs), and cell densities (a) during low and high tide conditions. Horizontal lines mark the threshold for salinity (blue solid line) (30) and temperature (red dashed line) (17 °C) values inducing cysts development. Values below the lines prevent cyst germination.

**Figure 3 toxins-16-00280-f003:**
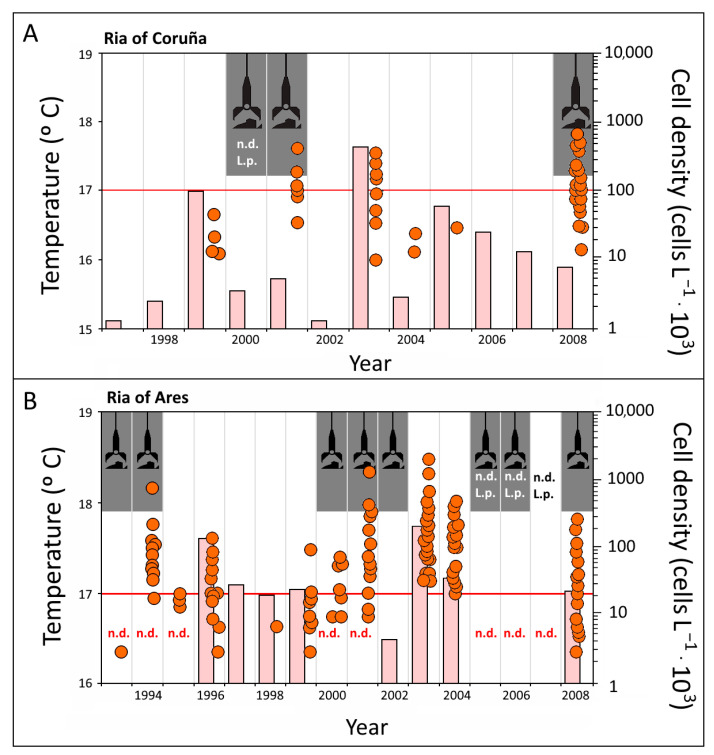
*L. polyedra* abundances (orange circles), bottom temperature at 10–15 m depth (red bars) and summer dredging activities (grey stripes with a black dredge icon) in the rias of (**A**) Coruña (1997–2008) and (**B**) Ares (1993–2008). The threshold temperature (17 °C) triggering cysts germination is indicated by a red line. No data are available for temperature (n.d.) or *L. polyedra* abundances (n.d. L.p.); these are also indicated.

**Figure 4 toxins-16-00280-f004:**
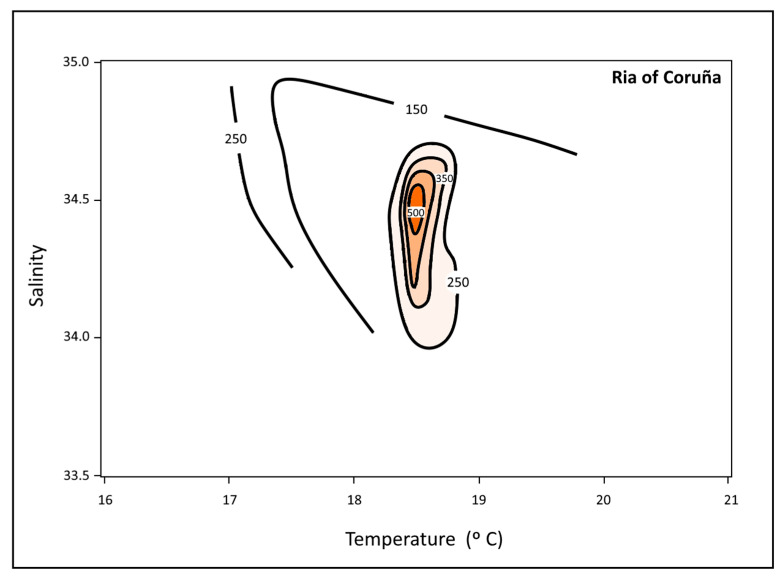
Window for temperature and salinity of *L. polyedra* abundance in the Ria of Coruña during the summer of 2008. The first cells develop at 17 °C, showing an optimum development of approximately 18.5 °C and 34.5 of salinity.

**Table 1 toxins-16-00280-t001:** Dredging activities inside the rias of Coruña and Ares. Sediments (m^3^) were retrieved in summer from the harbors of Coruña, Ares, and Sada and their nearby locations.

Ria	Year	Sand and Mud (m^3^)	Gravel and Rock (m^3^)
Coruña	2000	48,000	0
	2001	150,000	34,855
	2008	10,000	1000
Ares	1993–1994	26,000	377,000
	2000	80,000	0
	2001	34,000	25,000
	2002	16,500	13,600
	2005–2006	22,000	120
	2008	4000	0

**Table 2 toxins-16-00280-t002:** Bloom events of *L. polyedra* observed in the rias from 1993 to 2008. Proliferations due to anthropogenic (dredging), or natural (t is temperature; t_w_ is bottom seawater temperature) causes are also indicated as percentages.

Observed Red Tides	Ria of Coruña	Ria of Ares
Total cases	42	142
Explained areas	39 (92%)	118 (83%)
Dredging reasons (t < 17 °C)	30 (71%) *	63 (44%) *
Bottom temperature reasons (t_w_ > 17 °C)	9 (21%) *	55 (39%) *

* Significant correlation between *L. polyedra* blooms with dredging activities (*p* = 0.670; *p*-value < 0.05) and t_w_ > 17 °C (*p* = 0.580; *p*-value < 0.05).

## Data Availability

The data presented in this study are available in article.

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
