# Peer review of "Naturally and Anthropogenically Induced Lingulodinium polyedra Dinoflagellate Red Tides in the Galician Rias (NW Iberian Peninsula)"

_toxins, 2024, doi:10.3390/toxins16060280_

Round 1
Reviewer 1 Report
Comments and Suggestions for Authors
The paper is well written and presents interesting results about the influence of temperature and sediment resuspension on the onset of Lingulodinium polyedra blooms.
I really appreciate the Key Contribution that places emphasis on the bottom-up regulation of planktonic blooms. Usually only nutrients, temperature and hydrodynamic in the water column are used to explain these phenomena, while the role of the benthic dormant stages is ignored.
I have only little remarks:
Keywords: cysts is repeated two times
line 62 Maybe the right citation is Sobrino
line 66 The right citation should be [24]
line 218 water temperatures ranged from 14.5.....
line 302 in my opinion a reference should be reported after "with dredging operations"
line 371 maybe the right reference is: Figueroa, R.I., Bravo, I., 2005. Sexual reproduction and two different encystment strategies of Lingulodinium polyedrum (Dinophyceae) in culture. Journal of Phycology 41, 370–379. https://doi.org/10.1111/j.1529-8817.2005.04150.x
line 385 ref [56] here is useless
lines 372-379 Actually the dredging activities, differently from bioturbation, resuspend not only surface sediments where are newly produced cysts, not able to germinate because in the obligate dormancy period, but also cysts in deeper layers that have concluded the mandatory period of dormancy, so the single summer pulse should have another explanation. In fact, in the rias of Corona and Area, the blooms are continuous; in my opinion this part should be better explained
line 415 from table 2 I understand the the number of blooms caused by high bottom temperatures are 55 vs 9, so much more then double; maybe the % has been considered
Author Response
REVIEWER 1
Comments and Suggestions for Authors
The paper is well written and presents interesting results about the influence of temperature and sediment resuspension on the onset of Lingulodinium polyedra blooms.
I really appreciate the Key Contribution that places emphasis on the bottom-up regulation of planktonic blooms. Usually only nutrients, temperature and hydrodynamic in the water column are used to explain these phenomena, while the role of the benthic dormant stages is ignored.
Thanks for your kind words, we have put a lot of effort in the manuscript. Also, thank you for your comments and suggestions, they have contributed to improve the manuscript.
I have only little remarks:
Keywords: cysts is repeated two times
Line 39-40 (Keywords): We deleted `cyst´ duplicate
line 62 Maybe the right citation is Sobrino
Line 61: We add the cite of Sobrino (and renumbered the reference list)
line 66 The right citation should be [24]
Line 77: We correct and replace the citation by [24]
line 218 water temperatures ranged from 14.5.....
Line 218: We replace the sentence by ` water temperatures ranged from 14.5´
line 302 in my opinion a reference should be reported after "with dredging operations"
Line 303: we included two references about the dredging operations [48, 49]
- Spanish State Ports 2001. Investments, Chapter 5. In Statistical Yearbook; State Ports: Madrid, Spain, 2011; pp. 233-262
- Mouzo, E. Arrival of the ships that will participate in the integral dredging of the port of La Coruña. La Voz de Galicia (newspaper, digital edition) (20/02/2021): https://www.lavozdegalicia.es/noticia/coruna/2001/02/20/llegan-barcos-participaran-dragado-integral-puerto-corunes/0003_431786.htm (accesed on 01/06/2024).
line 371 maybe the right reference is: Figueroa, R.I., Bravo, I., 2005. Sexual reproduction and two different encystment strategies of Lingulodinium polyedrum (Dinophyceae) in culture. Journal of Phycology 41, 370–379. https://doi.org/10.1111/j.1529-8817.2005.04150.x
Line 372 (and Lines 710-711): Right, thanks, we have corrected the reference:
- Figueroa, R.I., Bravo, I. Sexual reproduction and two different encystment strategies of Lingulodinium polyedrum (Di-nophyceae) in culture. J. Phycol. 2005, 41, 370-379. https://doi.org/10.1111/j.1529-8817.2005.04150.x
line 385 ref [56] here is useless
Line 393: We delete the number of citation `Margalef considered red tides to be anomalies ..´
lines 372-379 Actually the dredging activities, differently from bioturbation, resuspend not only surface sediments where are newly produced cysts, not able to germinate because in the obligate dormancy period, but also cysts in deeper layers that have concluded the mandatory period of dormancy, so the single summer pulse should have another explanation. In fact, in the rias of Corona and Area, the blooms are continuous; in my opinion this part should be better explained
Line 378-383: We modified the sentence to better explained the case ` The dredging activities, differently from bioturbation, resuspended not only surface sediments where are newly produced cysts, not able to germinate because in the obligate dormancy period favoured by anoxic bottom conditions [12], but also cysts in deeper layers that have concluded the mandatory period of dormancy.are exposed to the water column, germinating under favourable conditions or sedimenting in the upper layers of the bottom more exposed to resuspension in other events.
line 415 from table 2 I understand the the number of blooms caused by high bottom temperatures are 55 vs 9, so much more then double; maybe the % has been considered
Lines 437-438: Right, thanks, we correct the sentence `…. high bottom temperatures is six times higher than in the Ria of Coruña (Table 2).´.

Reviewer 2 Report
Comments and Suggestions for Authors
One of the first major concerns was that there is a review paper, a 2024 one, reporting the HABs in the same region.
- DOI:
- 10.1039/D3EM00296A
The results need to be correlated with the above mentioned paper. How unique is the current study? this needs to be mentioned and highlighted in the text.
Table 1 has items missing and line numbers merged with the table, format and check
MDPI format has not been followed
The discussion could benefit from more discussions. The results are merely presented, add a critical comparison in the discussion section. Conclusions are conclusions, cut short and present briefly.
Figure legends need to be self explanatory.
Update references with recent ones.
Comments on the Quality of English Language
The language is fairly good.
Author Response
REVIEWER 2
Thank you for your comments and suggestions which have helped us to improve the manuscript.
One of the first major concerns was that there is a review paper, a 2024 one, reporting the HABs in the same region.
- DOI: 10.1039/D3EM00296A
The results need to be correlated with the above mentioned paper. How unique is the current study? this needs to be mentioned and highlighted in the text.
Right. We had included this paper in the references:
Lines 635-637: 20. Rodríguez, F.; Escalera, L.; Reguera, B.; Nogueira, E.; Bode, A.; Ruiz-Villarreal, M.; Rossignoli, A.E.; Ben-Gigirey, B.; Rey, V.; Fraga S. Red tides in the Galician rias: historical overview, ecological impact, and future monitoring strategies. Environ. Sci-Proc. Imp. 2023, 26, 16-34. https://doi.org/10.1039/D3EM00296A
Lines 413-422: We had used this reference in the Introduction section. However, it is a very relevant reference and we have included a discussion paragraph commenting on our data regarding the available historical series of L. polyedra blooms.
` Recently, Rodríguez et al. [20] have established that L. polyedra blooms were re-stricted to the Rías Baixas, the Rías de Vigo and Pontevedra, until the middle of the 20th century from the historical record of red tides on the coasts of Galicia (1916-2011). However, the data provided in this study show a higher recurrence of L. polyedra blooms in the Rias de Ares (1993-2008), Coruña (1999-2008) and Barqueiro (2012) (Fig. 2 and 3), as show the most recent historical record (2003-2011) [20]. L. polyedra blooms in the his-torical record [20], as well as the data provided in this study (Fig. 2 and 3), show a strong seasonality with a higher recurrence in spring-summer, with the exception of autumn of 2003 in the Ria de Ares. After the period of this study (1993-2008) only one L. polyedra bloom has been detected in the historical record in the Ares estuary in spring-summer 2011 [20].´
Table 1 has items missing and line numbers merged with the table, format and check MDPI format has not been followed
Right. We correct the Table 1 and use the MDPI format.
The discussion could benefit from more discussions. The results are merely presented, add a critical comparison in the discussion section. Conclusions are conclusions, cut short and present briefly.
We have included several more paragraphs in the discussion to improve the comparison of the data with available scientific information:
Section Discussion
Lines 377-382: ` Contrary to bioturbation, which is usually restricted to the uppermost sediments, where most of the cysts are still in their obligate dormancy condition, dredging activities affect the deeper layers rich in cysts with a potential to germinate [12]. This mechanism would explain the shorter (bioturbation in the case of Ría of Barqueiro) versus extended (dredging in the case of the rias of Ares and Coruña) blooms in the area.´
Lines 412-421: `Recently, Rodríguez et al. [20] have established that L. polyedra blooms were re-stricted to the Rías Baixas, the Rías de Vigo and Pontevedra, until the middle of the 20th century from the historical record of red tides on the coasts of Galicia (1916-2011). However, the data provided in this study show a higher recurrence of L. polyedra blooms in the Rias de Ares (1993-2008), Coruña (1999-2008) and Barqueiro (2012) (Fig. 2 and 3), as show the most recent historical record (2003-2011) [20]. L. polyedra blooms in the his-torical record [20], as well as the data provided in this study (Fig. 2 and 3), show a strong seasonality with a higher recurrence in spring-summer, with the exception of autumn of 2003 in the Ria de Ares. After the period of this study (1993-2008) only one L. polyedra bloom has been detected in the historical record in the Ares estuary in spring-summer 2011 [20].´
Lines 452-454: `In the Ares estuary, the spatial distribution of L. polyedra cysts has been studied, concentrating in the two shallowest internal areas of the estuaries, below 10 m depth, areas of low hydrodynamism or hydrodynamic shade dominated by fine muds [64].´
Section Conclusions
Lines 519-529: The historical data on L. polyedra red tides in the Galician rias provides a critical foun-dation for future research and management efforts. By understanding the key factors of bottom-up regulation that triggered harmful algal blooms, such as sediment disturbances, temperature conditions, and cyst transport vectors, we can develop predictive models and early warning systems to anticipate and mitigate future events. This knowledge informs better environmental management practices, such as timing dredging activities to avoid critical periods and monitoring ecosystem health indicators. It also supports the creation of regulatory frameworks to reduce human impacts and guides targeted research to uncover new mitigation strategies. Ultimately, this data is essential for protecting fisheries, aquaculture and the local economy, ensuring the sustainability of these vital resources.
Figure legends need to be self explanatory.
We have improved the figure footnotes by including the description of abbreviations and improving the display of graphs (Figure 1, 2 and 3) and we have corrected and changed the format of the tables 1 and 2.
Update references with recent ones.
We have corrected citations and included new citations by renumbering the bibliographic list correctly, and we have also revised the citations throughout the main text.

Reviewer 3 Report
Comments and Suggestions for Authors
This MS analyzed the causes responsible for Lingulodinium polyedra blooms in the coasts of the Iberian Peninsula and found that temperature and anthropogenical dredging contribute a significant role on it. This MS provide a plenty of data, especially the historical data, which is useful for those who interested in this species. However, some issues need to be clarified or revised:
(1) We know that cyst is very important for bloom initiation, but there is no direct evidence to prove that cyst is responsible for Lingulodinium polyedra bloom;
(2) Please combine the section of "study area" and "Materials and methods";
(3) Please provide more details on how to analyze the historical data (1993-2008)
Author Response
REVIEWER3
Even though Lingulodinium dinoflagellate blooms are not known to pose a direct human health risk, they can pose a risk to the aquaculture industry such as abalone, and their high visibility as red tides raises public concern. This work analyses a valuable long-term 1993-2008 data set from 3 Galician rias: Coruna Harbor, Ria of Ares, Rio of Barqueiro. Long term data (Fig.3) were only available for the first 2 locations, but during more detailed observations in July-Sept 2008 blooms were sparsest in the more open Barqueiro and densest in Ares. The 2008 results are used to define a 17C seasonal temperature window for blooms, but what is missing is details on bottom topography of the 3 areas and notably how this reflects differences in the build up of water column stratification (being the combined results of T and S). Are any fulll T and S depth profiles available? {line 206 mentions a vertical thermal gradient}. Most interestingly, but also least convincing, more dredging has occurred in Ares (8 dredgings) than Coruna (3 dredgings) , and sometimes dredging coincided with red tides (line 287), but then in other years red tides occurred without dredging (eg. Ares 1996,1999,2003, 2004). No dredging occurred in Barqueiro. The authors conclude that temperature-sediment disturbance can explain most red tides, but admit that diatom dominated yrs did not develop red tides (turbulent upwelling not allowing water column stratification?) and that upwelling affected Ares in a different way (line 137). To conclude in Table 2 that dredging explained 71% of blooms in Coruna and 44% of blooms in Ares is strongly biased by differences in bloom and dredging frequency! (try some statistics!)
First of all, thank you for your comments and suggestions have contributed decisively to the improvement of the manuscript.
It should also be noted that the most complete data on L. polyedra blooms in the Artabro Gulf and Barqueiro are our own.
We include statistic analysis of correlation between L. polyedra cell density with temperature and dredging activities periods to confirm the interrelationship between these parameters:
Line 293-296 (section Results): We include `In addition, there is a significant correlation between L. polyedra cell densities with dredging activities (P = 0.670; p-value < 0.05) and temperature of water in bottom > 17 °C (P = 0.580; p-value < 0.05) in both rias´
Lines 445-446: We include in the Table 2 (line 441) `*Significant correlation between L. polyedra blooms with dredging activities (P = 0.670; p-value < 0.05) and tw > 17 °C (P = 0.580; p-value < 0.05).´
Lines 578-581: We include this sentence in the section Material & Methods `The relationship between the dredging activities, temperature and cell density of L. polyedra in the rias was explored using Pearson correlation analysis (P), and by applying Student’s t-test to determine the level of statistical significance of correlations between parameters. The statistical analyses were performed using XLSTAT [75].´
Line 761-762: We include new reference:
- XLSTAT. Data Analysis and Statistical Solution for Microsoft Excel Addisonf (XLSTAT Version 2022.1); XLSTAT: Paris, France, 2022; Available online: https://www.xlstat.com/ (accessed on 1 June 2024).
Two important bits of data and arguments are missing
- A better , more complete explanation is needed to explain the differences in bloom behaviour of the 3 areas, in terms of depth contours (add into Fig.1), current regimes/upwellings, and water column stratification.
Right. We agree, we include the depth contours in the Figure 1 (line). The current regimes/upwellings and water column stratification were included in the discussion along the main text:
Lines 318-323: We include `Thermohaline data from the Ria of Coruña have been used to define an environmental temperature and salinity window for the presence and blooms of L. polyedra (Fig. 4). Red tides developed when temperature exceeded 17 °C, reaching maximum cell abundances at temperatures between 18 and 19 °C, and salinity varied between 34 and 35. Salinity showed a narrower variability during the summer due to the very low river flow to this ria (Varela and Prego, 2003).´
Lines 323-326: Also, we replace ` Temperature and salinity data were not available for the Ria of Ares in summer 2008; however, when considering all the available data from other years, the results were very similar to those from the ria of Coruña, even though the threshold for temperature increased up to 19.5 °C and salinity dropped down to 33.5´ by `Temperature and salinity data available for the Sts. 1-4 (Fig. 1) in the Ria of Ares in summer 2008 (INTECMAR, 2008) were similar to the Ria of Coruña, presenting salinities and temperatures higher than 34.0 and 17 °C, respectively. The same occurs to the Ria of Barqueiro (Ospina-Álvarez et al., 2010). ´
Lines 694: We include another reference: [45] INTECMAR, 2008. Galician Oceanographic Anual Report 2008. Trisquel Ediciones, 429 pp. ISBN: 978-84-613-3100-0
- More concerning, "no data on cysts" are available from Coruna (line 308) and Barqueiro, but there are data from Ares by Juan Blanco (but not presented here). Critically, are these viable cysts? To make the case that dredging stimulates blooms, the dredgings (is this anoxic mud?) need to have been checked for cysts (not done) and , much more challenging, the case should be made that dredging added to blooms already under development.
In conclusion, the data gaps that exist need to be clearly spelled out, and what is a proven fact and just speculation better separated.
Thanks for the comment, we have tried to include more information on L. polyedra cysts, especially at regional level, to make the narrative clearer for the reader. There are several references that study cyst germination and document the relationship of dinoflagellate blooms, including L. polyedra, to the cyst bank in the sediment, see [12-16, 26, 52-54]:
Lines 61-67: ` This dinoflagellate is considered a cosmopolitan warm-water species and has been recorded in many parts of the world [12]. However, it should be understood as a species restricted to warm temperate to tropical coasts only. Therefore, at the worldwide coastal scale, cyst germination and eventual blooms would depend on water temperature [13-16]. Interestingly, L. polyedra has become a highly frequent bloom-forming species in the temperate European Atlantic coasts, such as those of the Iberian Peninsula, in recent years [16-19].´
Lines 326-333: `Blanco [13] suggested that temperature controls the rate of cyst germination, while Peña-Manjarrez [52] pointed out that when water temperature increases to approximately 17 °C, a “thermal window” is reached for this species, promoting cyst germination, together with a decrease of their relative abundance in surface sediments. This role of temperature was also proposed by other authors within a temperature range from 17 to 23 °C [15, 52-54]. Maximum threshold temperature may change according to geo-graphical area, but the minimum temperature reported for cyst germination is always 17 °C.´
We include more discussion about this question with new references. It is also mentioned in the text and the following references:
Line 625 & 741-742: [13] Blanco, J., 1990. Cyst germination of two dinoflagellate species from Galicia (NW Spain). Sci. Mar. 54, 287-291 and (new reference) [67] Blanco, J. 1988. Distribución vertical y asociación al sedimento de los quistes de dinoflagelados en la ría de Ares y Betanzos. Investigación Pesquera, 52(3): 335-344 (both in Spanish):
Lines 459-467: We include in the main text `L.p. cysts that were stored in anoxic conditions had much higher germination percentages than those stored in oxic environment, which practically does not germinate [13]. In this way, the cysts buried deeper in the sediment make up important and effective reserve populations for motile phases (Blanco, 1990). In the Sada coast (St. 3 in Fig. 1) of the Ria of Ares, Blanco [67] observed that the vertical distribution on L.p. in the sediment shows one or more subsurface maxima. In this way, the dredging activity can remove mud sediments at a greater depth, i.e. usually anoxic. There, a specie as L.p., which cysts are very resistant, are accumulate as permanent inoculum [67].´
Line 649: [25] Blanco, J., 1989. Quistes de dinoflagelados de las costas de Galicia. I. Dinoflagelados Gonyaulacoides. Sci. Mar., 53: 785-796. (see page 793):
(See page 793): “The correlation between the typical form of this cyst and the mobile phase has been established on several occasions (Blanco, 1989 and the references cited therein). “
Line 734 (new reference): [64] Blanco, J. 1989. Distribución de quistes de dinoflagelados en la ría de Ares y Betanzos. Bol. IEO, 5 (2): 11-18.
Lines 467-468: We include in the main text `The highest cysts presence in sediment were in ria mud sediments, i.e. the Lorbé-Sada coast (St. 1-3 in Fig. 1) with contents of 4-20 cysts·L-1 of sediment (Blanco, 1989).´
In detail:
Title: the taxonomic detail of Stein/Dodge is not relevant to a Toxins auduence, but perhaps refer to "dinoflagellate red tides" instead
Lines 3-4: Thanks, we agree. We change the title: Natural and anthropogenically-induced Lingulodinium polyedra dinoflagellate red tides in the Galician rias (NW Iberian Peninsula)
line 7. on the coasts
Line 18: We included `on the coasts´.
line 14. thermal window resulting from seasonal window=seasonal temperature window
Line 25: Ok, we agree. We replace by ` a “seasonal thermal window” conditions ´.
line 19. cyst
Lines 39-40 (Keyword): We delete the repeated term `cyst´.
line 23. latency period=lag period
Line 34: We replace `latency period´ by `lag period´.
line 51. Cosmopolitan=cosmopolitan warm-water species
Line 62: We replace `cosmopolitan´ by `cosmopolitan warm-water´.
line 77. resistance stage=resistant resting stage
Line 88: We replace ` resistance stage´ by `resistant resting stage ´.
line 88. the old claim that Lingulodinium produces saxitoxin is no longer correct; this was based on positive intraperitoneal mouse bioassays only, not analytical chemistry. The precise human oral potency of yessotoxin is not known, but yessotoxin-generating blooms have been associated with abalone mortality eg in South Africa
Line 99-100: Right, thanks. We agree, we have rewritten the sentence: `In several parts of the world, L. polyedra has been associated with the production of paralytic toxins, such as yessotoxins [31, 32], and ichthyotoxins [33].´
Fig.2. Define in the fig legend where the S and T threshhold for cyst development come from.
Lines 237, 238-243: We modified the Fig. 2C included the threshold data and included the information in the legend: `(C) Ria of Barqueiro showing temperature (t), salinity (s), current speed (cs) and cell densities (a) during low and high tide conditions. Horizontal lines mark the threshold for salinity (blue solid line) (30) and temperature (red dashed line) (17 °C) values inducing cysts development. Values below the lines, prevent cyst germination.´
Fig.3. What is portrayed as the grey images representing dredging are not clear. Use a different icon.
Line 270: Ok, we replace the icon to improve the display of graphics.
line 573. Lingulodinium polyedra in italics
Line 575: Ok, we correct this, and along the main text.

Reviewer 4 Report
Comments and Suggestions for Authors
Even though Lingulodinium dinoflagellate blooms are not known to pose a direct human health risk, they can pose a risk to the aquaculture industry such as abalone, and their high visibility as red tides raises public concern. This work analyses a valuable long-term 1993-2008 data set from 3 Galician rias: Coruna Harbor, Ria of Ares, Rio of Barqueiro. Long term data (Fig.3) were only available for the first 2 locations, but during more detailed observations in July-Sept 2008 blooms were sparsest in the more open Barqueiro and densest in Ares. The 2008 results are used to define a 17C seasonal temperature window for blooms, but what is missing is details on bottom topography of the 3 areas and notably how this reflects differences in the build up of water column stratification (being the combined results of T and S). Are any fulll T and S depth profiles available? {line 206 mentions a vertical thermal gradient}. Most interestingly, but also least convincing, more dredging has occurred in Ares (8 dredgings) than Coruna (3 dredgings) , and sometimes dredging coincided with red tides (line 287), but then in other years red tides occurred without dredging (eg. Ares 1996,1999,2003, 2004). No dredging occurred in Barqueiro. The authors conclude that temperature-sediment disturbance can explain most red tides, but admit that diatom dominated yrs did not develop red tides (turbulent upwelling not allowing water column stratification?) and that upwelling affected Ares in a different way (line 137). To conclude in Table 2 that dredging explained 71% of blooms in Coruna and 44% of blooms in Ares is strongly biased by differences in bloom and dredging frequency! (try some statistics!)
Two important bits of data and arguments are missing
1. A better , more complete explanation is needed to explain the differences in bloom behaviour of the 3 areas, in terms of depth contours (add into Fig.1), current regimes/upwellings, and water column stratification.
2. More concerning, "no data on cysts" are available from Coruna (line 308) and Barqueiro, but there are data from Ares by Juan Blanco (but not presented here). Critically, are these viable cysts? To make the case that dredging stimulates blooms, the dredgings (is this anoxic mud?) need to have been checked for cysts (not done) and , much more challenging, the case should be made that dredging added to blooms already under development.
In conclusion, the data gaps that exist need to be clearly spelled out, and what is a proven fact and just speculation better separated.
In detail:
Title: the taxonomic detail of Stein/Dodge is not relevant to a Toxins auduence, but perhaps refer to "dinoflagellate red tides" instead
line 7. on the coasts
line 14. thermal window resulting from seasonal window=seasonal temperature window
line 19. cyst
line 23. latency period=lag period
line 51. Cosmopolitan=cosmopolitan warm-water species
line 77. resistance stage=resistant resting stage
line 88. the old claim that Lingulodinium produces saxitoxin is no longer correct; this was based on positive intraperitoneal mouse bioassays only, not analytical chemistry. The precise human oral potency of yessotoxin is not known, but yessotoxin-generating blooms have been associated with abalone mortality eg in South Africa
Fig.2. Define in the fig legend where the S and T threshhold for cyst development come from.
Fig.3. What is portrayed as the grey images representing dredging are not clear. Use a different icon.
line 573. Lingulodinium polyedra in italics
Comments on the Quality of English LanguageMinor improvements needed as indicated above
Author Response
REVIEWER 4
This MS analyzed the causes responsible for Lingulodinium polyedra blooms in the coasts of the Iberian Peninsula and found that temperature and anthropogenical dredging contribute a significant role on it. This MS provide a plenty of data, especially the historical data, which is useful for those who interested in this species. However, some issues need to be clarified or revised:
Thank you for your insight into the manuscript, we have invested a lot of effort in trying to show interesting, lesser known aspects of the blooms of L. polyedra. Your comments and suggestions have contributed to the improvement of the manuscript.
(1) We know that cyst is very important for bloom initiation, but there is no direct evidence to prove that cyst is responsible for Lingulodinium polyedra bloom;
Thanks for the comment, we have tried to include more information on L. polyedra cysts, especially at regional level, to make the narrative clearer for the reader. There are several references that study cyst germination and document the relationship of dinoflagellate blooms, including L. polyedra, to the cyst bank in the sediment, see [12-16, 26, 52-54]:
Lines 61-67: ` This dinoflagellate is considered a cosmopolitan warm-water species and has been recorded in many parts of the world [12]. However, it should be understood as a species restricted to warm temperate to tropical coasts only. Therefore, at the worldwide coastal scale, cyst germination and eventual blooms would depend on water temperature [13-16]. Interestingly, L. polyedra has become a highly frequent bloom-forming species in the temperate European Atlantic coasts, such as those of the Iberian Peninsula, in recent years [16-19].´
Lines 326-333: `Blanco [13] suggested that temperature controls the rate of cyst germination, while Peña-Manjarrez [52] pointed out that when water temperature increases to approximately 17 °C, a “thermal window” is reached for this species, promoting cyst germination, together with a decrease of their relative abundance in surface sediments. This role of temperature was also proposed by other authors within a temperature range from 17 to 23 °C [15, 52-54]. Maximum threshold temperature may change according to geo-graphical area, but the minimum temperature reported for cyst germination is always 17 °C.´
We include more discussion about this question with new references. It is also mentioned in the text and the following references:
Line 625 & 741-742: [13] Blanco, J., 1990. Cyst germination of two dinoflagellate species from Galicia (NW Spain). Sci. Mar. 54, 287-291 and (new reference) [67] Blanco, J. 1988. Distribución vertical y asociación al sedimento de los quistes de dinoflagelados en la ría de Ares y Betanzos. Investigación Pesquera, 52(3): 335-344 (both in Spanish):
Lines 459-467: We include in the main text `L.p. cysts that were stored in anoxic conditions had much higher germination percentages than those stored in oxic environment, which practically does not germinate (Blanco, 1990). In this way, the cysts buried deeper in the sediment make up important and effective reserve populations for motile phases (Blanco, 1990). In the Sada coast (St. 3 in Fig. 1) of the Ria of Ares, Blanco (1988) observed that the vertical distribution on L.p. in the sediment shows one or more subsurface maxima. In this way, the dredging activity can remove mud sediments at a greater depth, i.e. usually anoxic. There, a specie as L.p., which cysts are very resistant, are accumulate as permanent inoculum (Blanco, 1988).´
Line 649: [25] Blanco, J., 1989. Quistes de dinoflagelados de las costas de Galicia. I. Dinoflagelados Gonyaulacoides. Sci. Mar., 53: 785-796. (see page 793):
(See page 793): “The correlation between the typical form of this cyst and the mobile phase has been established on several occasions (Blanco, 1989 and the references cited therein). “
Line 734 (new reference): [64] Blanco, J. 1989. Distribución de quistes de dinoflagelados en la ría de Ares y Betanzos. Bol. IEO, 5 (2): 11-18.
Lines 467-468: We include in the main text `The highest cysts presence in sediment were in ria mud sediments, i.e. the Lorbé-Sada coast (St. 1-3 in Fig. 1) with contents of 4-20 cysts·L-1 of sediment (Blanco, 1989).´
(2) Please combine the section of "study area" and "Materials and methods";
Thank you for the suggestion, however, due to the format of the journal where the Material and methods section goes at the end, we believe it is necessary to keep the Study area section before the results to put the reader in the regional context.
(3) Please provide more details on how to analyses the historical data (1993-2008)
Thanks, we agree. We include more information about historical data, also comparing with Rodriguez et al. [20], in section Discussion:
Lines 413-422: `Recently, Rodríguez et al. [20] have established that L. polyedra blooms were re-stricted to the Rías Baixas, the Rías de Vigo and Pontevedra, until the middle of the 20th century from the historical record of red tides on the coasts of Galicia (1916-2011). However, the data provided in this study show a higher recurrence of L. polyedra blooms in the Rias de Ares (1993-2008), Coruña (1999-2008) and Barqueiro (2012) (Fig. 2 and 3), as show the most recent historical record (2003-2011) [20]. L. polyedra blooms in the his-torical record [20], as well as the data provided in this study (Fig. 2 and 3), show a strong seasonality with a higher recurrence in spring-summer, with the exception of autumn of 2003 in the Ria de Ares. After the period of this study (1993-2008) only one L. polyedra bloom has been detected in the historical record in the Ares estuary in spring-summer 2011 [20].´

Round 2
Reviewer 4 Report
Comments and Suggestions for Authors
I am happy with the authors responses to reviewer comments, notably the inclusion of more discussion on sediment cysts